# Neural Time Integrator with Stage Correction

## Abstract

Numerical simulation of dynamical systems requires time integration solvers that balance accuracy and computational efficiency. Recent work indicates that neural integrators, a hybrid of classical numerical integration and machine learning, can achieve significant performance gains. Building upon this idea, we propose a new type of neural integrator that introduces stage corrections inspired by the fact that traditional time integration schemes such as Runge-Kutta exhibit different error characteristics at each stage. Specifically, our method corrects numerical errors immediately after each stage evaluation by using a neural network, mitigating error propagation across stages. This enables the use of larger time steps while preserving stability and accuracy. We demonstrate that our approach is at least one order of magnitude more accurate than existing hybrid methods for complex nonlinear dynamical systems when integrated with the same step size.

## 1 Introduction

Accurate numerical simulations of dynamical systems are critical in fields ranging from computational fluid dynamics to computational physics and from material science to chemical engineering. Traditionally, solving the governing equations for these systems relies on numerical discretization methods, which are computationally expensive and lead to trade-offs between speed and accuracy.

In recent years, machine learning (ML) algorithms have gained popularity due to their versatility and potential for reducing computational demands. These methods can generally be divided into two main categories. The first category is purely **data-driven**, where no explicit knowledge of the underlying physical laws is used. Examples include Fourier Neural Operators (FNO) Li et al. (2020) and DeepONets Lu et al. (2021). Some other works focused on data-driven dynamics discovery Pan & Duraisamy (2018); Liu et al. (2022); Chen & Xiu (2021). The second category is **physics-informed**, such as Physics-Informed Neural Networks (PINNs) Raissi et al. (2017; 2019); Rad et al. (2020), which enforce the governing equations and boundary conditions without requiring labeled data. While both approaches have successfully produced surrogate models that can replace traditional numerical solvers, they face limitations when applied to partial differential equations (PDEs) involving complex geometries and boundary conditions Lu et al. (2022); McGreivy & Hakim (2024), chaos Choudhary et al. (2020); Han et al. (2021); Greydanus et al. (2019), stiffness Huang & Leimkuhler (1997); Liang et al. (2022), and long-term prediction Wang et al. (2022).

More recently, hybrid approaches that combine machine learning with numerical simulations have emerged, offering a promising alternative for accelerating the simulation of dynamical systems. These methods leverage the speed of low-fidelity simulations while improving accuracy through error correction via deep neural networks (see Figure 1). For this reason, we term such methods *Neural Integrators*. As a result, neural integrators are data-driven, physics-informed, and fundamentally **simulation-based**. The integration of simulations confers two key advantages:

- **Enriched physical information and improved reliability.** By combining classic time-stepping methods with neural networks (NNs), these approaches retain the intrinsic physics of the system. Low-fidelity simulations provide reasonable approximations of system states, making it more feasible for NNs to learn and correct errors rather than to directly predict the solution.

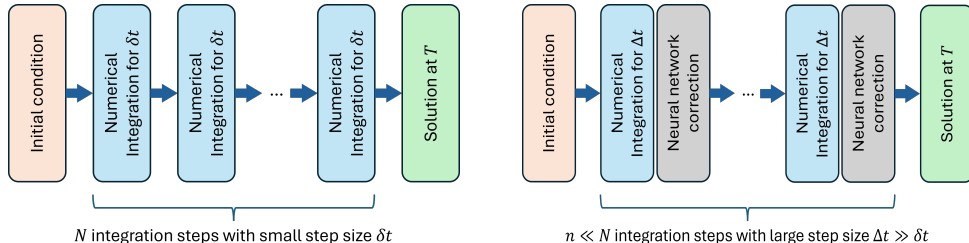

Figure 1: Illustration of the hybrid approach. **Left:** classical high-fidelity numerical simulation. **Right:** efficient high-fidelity simulation with network corrections.

- **Seamless integration with existing simulation codes.** Unlike many ML models that aim to fully replace traditional simulations, hybrid approaches allow for the incorporation of neural networks into existing simulation frameworks. By modifying only the time-stepping components, these methods can achieve the same scalability as the original simulation code, making them particularly suitable for complex large-scale HPC applications such as climate models.

A notable example of this neural integrator paradigm is NeurVec Huang et al. (2023) (see Figure 2 (a)), which has demonstrated speedups of tens to hundreds of times compared to traditional solvers. However, the theory of NeurVec was developed for the Euler method, and NeurVec yields suboptimal results for multistage time integration methods. In this work, we propose a new type of neural time integrator based on a stage correction strategy (NeurTISC) and demonstrate its advantages over NeurVec. In our method, we correct numerical errors immediately after each stage evaluation by using a neural network (see Figure 2 (b)). This allows us to adopt a large step size when using a classical integrator and at the same time mitigate error propagation across stages.

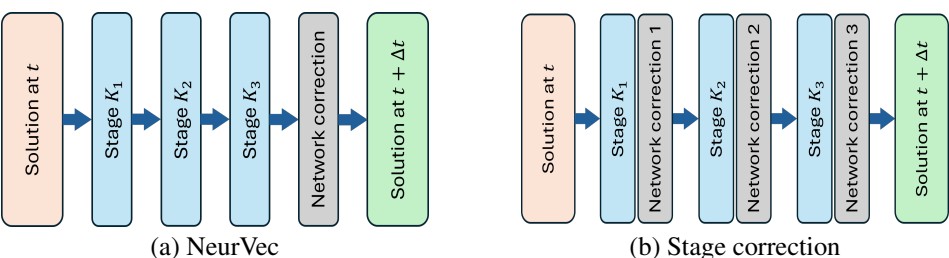

(a) NeurVec  (b) Stage correction

Figure 2: Comparison between NeurVec and Stage correction proposed in this paper. The figure demonstrates the difference when applied to a 3-stage numerical time stepper, e.g., Runge-Kutta 3.

We provide empirical validation of this strategy, demonstrating its ability to significantly improve performance in stiff dynamical system simulations. Our experimental results show that the stage correction strategy improves the accuracy of at least one order of magnitude compared to NeurVec when both models have the same number of parameters and are integrated with the same time step size. We also demonstrate that the stage correction strategy possesses stronger error correction power so that it can integrate the system stably for a longer time period compared to NeurVec. Lastly, we show through a benchmark that using a trainable rational activation function Telgarsky (2017); Boullé et al. (2020) in the correction network significantly improves error correction and accuracy in longer time integration compared to classic activation functions like GeLU and $\tanh$.

## 2 BACKGROUND

**Classic Solvers for Dynamical Systems.** A time-dependent dynamical system is typically depicted by a system of ODEs

$$\frac{du(t)}{dt} = F(u(t)), \quad u(0) = u_0 \in \mathbb{R}^d .$$

(1)

where $t \in [t_0, t_f]$ and $u(t) \in \mathbb{R}^d$ is the system state. To solve this system numerically, one can employ a time integration method that propagates $u(t)$ by an iterative formula

$$u_{n+1} = u_n + \mathbb{S}(F, u_n, \delta t), \quad n = 0, 1, \ldots, \tag{2}$$

where $u_n$ is the short hand for $u(t_n)$, $\delta t = t_{n+1} - t_n$ is the step size, and $\mathbb{S}$ defines a specific integration scheme so that

$$\mathbb{S}(F, u_n, \delta t) \approx \int_{t_n}^{t_{n+1}} F\left(u(t_n + \tau)\right) d\tau. \tag{3}$$

For example, an $s$-stage explicit Runge-Kutta (ERK) method approximates the solution with

$$k_i = F\left(u_n + \delta t \cdot \sum_{j=1}^{i-1} a_{ij} k_j\right), \quad i = 1, \ldots, s,$$
$$u_{n+1} = u_n + \delta t \cdot \sum_{i=1}^{s} b_i k_i. \tag{4}$$

$\mathbb{S}$ for ERK is a linear summation of all the slopes $\{k_i, i = 1, \ldots, s\}$.

The ODEs may also arise from time-dependent PDE simulations, where the governing equation can also be written in the general form

$$\frac{\partial u}{\partial t} = G(x, u, \partial u/\partial x, \partial^2 u/\partial x^2, \ldots). \tag{5}$$

The right-hand side (RHS) function $G$ contains the spatial derivatives that are typically approximated with a spatial discretization method such as finite difference method and finite element method.

**The NeurVec Framework.** NeurVec is a NN-based corrector added to the estimated solution generated by an ODE solver. It allows the ODE solver to use a larger step size, typically beyond the stability limit, therefore accelerating the solution process significantly. Consider the iterative formula (2) with a $k-$times larger step size $\Delta t = k \cdot \delta t$, NeurVec learns a mapping from the state to the error correction and uses it to compensate the errors:

$$\hat{u}_{k(n+1)} = \hat{u}_{kn} + \mathbb{S}(F, \hat{u}_{kn}, k\delta t) + \text{NeurVec}(\hat{u}_{kn}, \Theta), \quad n = 0, 1, \ldots \tag{6}$$

The corrected solution $\hat{u}_{k(n+1)}$ intends to approach the solution with high accuracy. To understand the potential speedup, we take the ERK method (4) for an example. The ERK method with a fine step size has a computational complexity $O(1/\delta t)$. In contrast, the evaluation time for NeurVec is $O((1 + \epsilon)/(k\delta t))$ where $\epsilon$ is the runtime ratio between the NN inference time and the per-step cost of the ODE solver. As $\epsilon$ decreases, the speedup by using NeurVec increases, with an upper bound $k$.

## 3 NEURAL TIME INTEGRATOR WITH STAGE CORRECTION

In this section, We provide the motivation and detailed description of our neural time integrator with a stage-by-stage correction strategy.

**Motivation.** Our stage-by-stage correction strategy is inspired by the observation that the time integration errors at different stages are of different nature, in both magnitudes, smoothness, and frequency, as shown in Figure 3. NeurVec trains a single NN to learn the mixed correction at the end of each time step. This is quite challenging, especially when one attempts to keep the NN architecture simple and the number of NN parameters low for best speedup.

**Correction at Intermediate Stages.** For the ease of illustration, we use the Heun's method (a second-order ERK) to demonstrate how the stage corrections are carried out in our approach. As shown below, NeurTISC modifies this scheme by adding to each $k_i$ stage an NN correction that takes $F(\cdot)$ as the input.

$$\begin{aligned} k_1 &= F(u_n), \\ k_2 &= F(u_n + k_1 \Delta t), \\ u_{n+1} &= u_n + \Delta t/2(k_1 + k_2). \end{aligned} \quad \Longrightarrow \quad \begin{aligned} \hat{k}_1 &= (\mathbb{I} + \text{NN}_1)(F(\hat{u})), \\ \hat{k}_2 &= (\mathbb{I} + \text{NN}_2)(F(\hat{u} + \hat{k}_1 \Delta t)), \\ \hat{u}_{n+1} &= u_n + \Delta t/2(\hat{k}_1 + \hat{k}_2). \end{aligned}$$

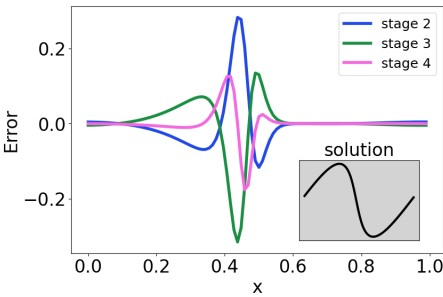
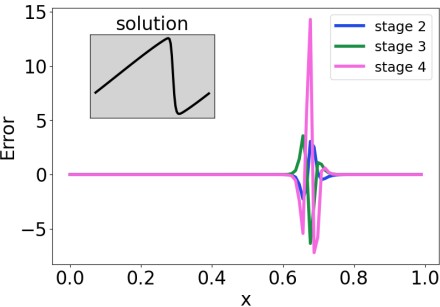

Figure 3: Stage errors when solving 1D Burgers equation with a large step size $\Delta t$. The errors are computed as $k_i - k_i^{\text{REF}}$ where the reference $k_i^{\text{REF}}$ are obtained with a very fine step size. **Left:** stage errors before the shock is formed. **Right:** stage errors under a sharp shock.

Here $\mathbb{I}$ is the identity map and $\text{NN}_i$ are a trained neural network to compensate the error for the $i-$th stage.

**Simple Neural Networks.** We employ a simple NN structure, consisting of two fully connected layers and a rational activation function. The activation function $\sigma$ is defined as

$$\sigma(x) = \frac{ax^3 + bx^2 + cx + d}{1 + (fx + g)^2}$$

where $a, b, c, d, f, g$ are learnable parameters. The parametrization of the denominator is used to avoid possible singularities. The rational activation function has been observed to outperform standard activation functions (e.g., ReLU) in tasks such as approximating solutions to PDEs Boullé et al. (2020); Huang et al. (2023). Noticeably, the advantage of rational activations is that they are trainable, resolving the issue of hypothetical selection of activation functions Raissi et al. (2019); Raissi (2018); Jagtap et al. (2020); Boullé et al. (2020). In addition, deep neural networks equipped with trainable rational activations have superior approximation power compared to ReLU activated networks Telgarsky (2017); Boullé et al. (2020), while maintaining a comparable bound on the generalization error Zhang & Kileel (2023). Following Boullé et al. (2020), we initialize the rational activation function such that it resembles the ReLU (GeLU) function when $x$ is close to the origin. The effectiveness of the rational activations is demonstrated in Section 4.

**Data Preparation.** For each initial state $u_0^{(j)}, j = 1, 2, \ldots, P$, we generate a sequence of reference solutions $\{u_{\text{ref}}^{(j)}(t_i) : i = 0, 1, \ldots, N_{\text{ref}}, j = 1, 2, \ldots, P\}$ by integrating the system with a sufficiently small time step $\delta t$. In order to train NeurTISC with time step $\Delta t = k\delta t, k \gg 1$, we extract training data $\{u^{(j)}(t_{ki}) = u_{\text{ref}}^{(j)}(t_{ki}) : i = 0, 1 \ldots, N_{\text{train}}, j = 1, 2, \ldots, P\}$ by subsampling $u_{\text{ref}}$ every $k$ time steps.

**Weighted Loss and multi-step training.** To produce corrections at a future time beyond the training interval, we integrate the model for multiple time steps and average the loos over the $L$ steps. Since the corrected solutions are used as the input for the next step, the prediction errors also accumulate during time integration. To alleviate this issue, we assigns decaying weights to predictions later in time. The weighted $\ell_2$ loss is thus

$$\ell(\Theta) = \frac{1}{Ld} \sum_{m=1}^{L} \beta^m \cdot \left\| u^{(j)}(t_\alpha + m\Delta t) - \hat{u}^{(j)}(t_\alpha + m\Delta t) \right\|^2$$

where $\alpha$ and $j$ are the batch dimensions, $\beta \in (0, 1]$ is a hyperparameter that controls the decaying speed. Our empirical experiences suggest that $\beta \sim 0.95$ can improve the robustness of the model training for long time windows (e.g. $L > 10$).

During training, we start training the model with a short time window e.g., $L = 2, 3$, and gradually increase the window size. Because models for a shorter path is easier to learn, in such a way the parameters are guided towards the optimum for longer paths step by step. This multi-phase training strategy significantly reduces the occurrences of blowing up the solution due to instability.

## 4 NUMERICAL EXPERIMENTS

To evaluate the performance of NeurTISC, we consider three dynamical systems of distinct nature – elastic pendulum, viscous Burgers equation, and Kuramoto–Sivashinsky (KS) equation. The first task is an ODE problem, the second is a classic PDE problem from fluid dynamics, and the third one is a nonlinear chaotic PDE problem. We extensively compare NeurTISC with the existing method NeurVec, demonstrating the improved accuracy and numerical stability, the generalization to different initial conditions, and the ability to handle unseen future data. We use the same training setup such as training/testing data, loss function and learning rate scheduler for both models. We use the loss as a measure of how accurate the prediction can be for different models.

**Elastic Pendulum.** The elastic pendulum problem describes the motion of a freely-swinging point mass connected to spring under gravity. The governing equation is given by

$$\begin{pmatrix} \ddot{\theta} \\ \ddot{r} \end{pmatrix} = \begin{pmatrix} \frac{1}{r}(-g\sin\theta - r\dot{\theta}) \\ r\dot{\theta}^2 - \frac{k}{m}(r - \ell_0) + g\cos\theta \end{pmatrix}. \tag{7}$$

Here $g$ is gravity constant, $k$ and $\ell_0$ are stiffness and relaxed length of the spring. The state of the system is described by $(r(t), \theta(t))$, the length of the spring and the angle between the spring and vertical direction. This system is known to be potentially chaotic Breitenberger & Mueller (1981).

To reduce to the standard form (1), we introduce dummy variables $\dot{r}$ and $\dot{\theta}$. We sample training paths by randomly sampling the initial state, with details listed in the table Table 1 below. We sample

| $r(0)$ | $\dot{r}(0)\ [s^{-1}]$ | $\theta(0)$ | $\dot{\theta}(0)\ [s^{-1}]$ |
|---|---|---|---|
| $\text{Unif}(3/4, 5/4)\ell_0$ | $\mathcal{N}(0, 0.1^2)\ell_0$ | $\text{Unif}(-\pi/8, \pi/8)$ | $\mathcal{N}(0, (0.1\pi)^2)$ |

Table 1: Initial condition of the system.

128 training paths and 32 test paths, all integrated with RK4 with fine time step $\delta t = 0.002$. We train NeurTISC and NeurVec using a coarse time step $\Delta t = 150\delta t$. The two neural integrators roughly have the same total number of parameters, with NeurVec having a hiddem dimension of 128 and NeurTISC having only 32. In Table 2, we summarize test accuracy of both models when integrated forward for different number of time steps. Figure 4 shows the predicted results of NeurTISC for

| L | 2 | 3 | 5 | 10 |
|---|---|---|---|---|
| RK4 no correction | 3.658 | 2.427e7 | Failure | Failure |
| NeurVec | 9.676e-2 | 3.223e4 | Failure | Failure |
| NeurTISC (ours) | **4.730e-4** | **1.756e-5** | **9.435e-6** | **1.908e-5** |

Table 2: Comparison of test accuracy on elastic pendulum. "Failure" indicates the solution blows up.

four different scenarios. In all these cases, the trajectories predicted by NeurTISC match the ground truth perfectly while the solution blows up when using NeurVec with the same step size.

Furthermore, we evaluate the effectiveness of the rational activation function. We test NeurTISC with rational, GeLU, and tanh activations and compare its performance. Table 3 shows that rational significantly outperform the other two options.

| L | 2 | 3 | 5 | 10 |
|---|---|---|---|---|
| GeLU | 2.923e-1 | 3.257e-2 | 8.903e-2 | Failure |
| tanh | 1.033e-3 | 1.782e-4 | 5.633e-5 | 1.004e-4 |
| rational | **4.730e-4** | **1.756e-5** | **9.435e-6** | **1.908e-5** |

Table 3: Comparison of test accuracy on elastic pendulum using NeurTISC with different activations. "Failure" indicates the solution blows up.

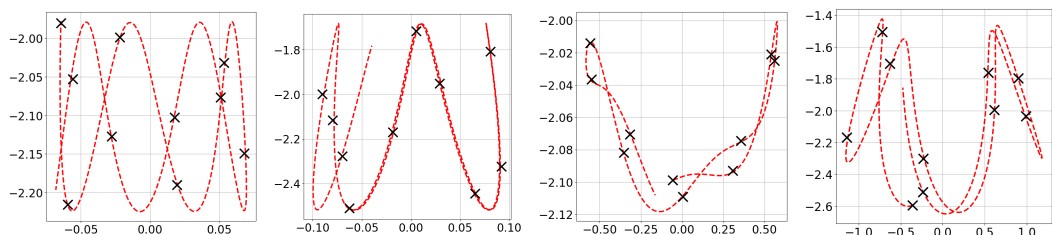

Figure 4: 10-step predicted solutions of NeurTISC on elastic pendulum. The red line is the exact path, the black crossings are NeurTISC outputs. From left to right, the paths correponds to initial conditions with (i) small radial and angular velocity, (ii) large radial velocity and small angular velocity, (iii) small radial velocity and large angular velocity, and (iv) large radial and angular velocity.

**Viscous Burgers Equation.** We consider the 1D viscous Burgers equation on $[0, 1]$ with periodic boundary condition

$$\frac{\partial u}{\partial t} = -u\frac{\partial u}{\partial x} + \nu\frac{\partial^2 u}{\partial x^2}. \tag{8}$$

To reduce to the ODE form (1), we use a uniform grid on $[0, 1]$ and use finite difference methods to approximate the spacial derivatives. To make the problem more challenging, we use the following sinusoidal initial condition to form a shock:

$$u(0, x) = a_1 + a_2 \sin(2\pi x) + a_3 \cos(2\pi x). \tag{9}$$

The coefficients $a_1, a_2, a_3$ are randomly sampled from a Gaussian distribution $\mathcal{N}(0.5, 0.1)$. The dissipation parameter is set to $\nu = 0.008$. The fine time step is $\delta t = 0.0001$ and the coarse time step for training is $\Delta t = 200\delta t$. The spatial grid has $n = 96$ equally spaced points. We choose this setting to make the solution develops a shock gradually and integration using the fine time step is stable, but integrating with the coarse step $\Delta t$ is unstable (see Figure 5 as an example).

We generate 64 training trajectories and 8 test trajectories. We start the training with $L = 2$ and extend to $L = 5$. The dimension of the hidden layer is 1024 for NeurTISC and 4096 for NeurVec so that the two models have approximately the same number of parameters. The test results are reported in Table 4. Figure 5 shows the predicted solutions for $L = 5$ steps using NeurTISC and using RK4 without correction.

| L | 2 | 5 |
|---|---|---|
| RK4 no correction | 2.384e-7 | 9.432e-3 |
| NeurVec | 2.322e-8 | 1.131e-3 |
| NeurTISC (Ours) | **2.804e-9** | **1.009e-4** |

Table 4: Comparison of test accuracy on viscous Burgers equation.

We also evaluate the effectiveness of rational activation following the previous experiment on the elastic pendulum problem. The results are shown in Table 5. The benefit of a trainable rational activation is marginal for a small window size $L = 2$. However, the benefit of rational activation becomes evident for a larger window size $L = 5$.

| L | 2 | 5 |
|---|---|---|
| GeLU | 3.441e-9 | 5.271e-4 |
| tanh | 3.083e-9 | 2.839e-4 |
| rational | **3.031e-9** | **1.627e-4** |

Table 5: Comparison of test accuracy on viscous Burgers equation using NeurTISC with different activations.

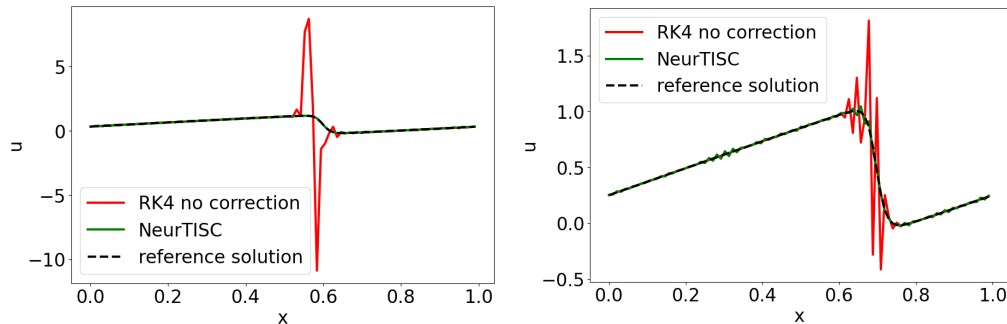

Figure 5: 5-step predicted solutions of NeurTISC applied to 1D viscous Burgers equation on $[0, 1]$. **Left:** the error of RK4 under a shock (note the different scaling in $y$-axis). **Right:** After the shock is smoothed by dissipation, RK4 still suffers from large high-frequency errors.

**Kuramoto–Sivashinsky Equation.** This is a 4th order nonlinear PDE that exhibits chaotic behaviour when interrogated for sufficiently long times. The governing equation we consider is

$$\frac{\partial u}{\partial t} = \frac{\partial^2 u}{\partial x^2} + \frac{\partial^4 u}{\partial x^4} + \frac{1}{2} \cdot \left( \frac{\partial u}{\partial x} \right)^2. \tag{10}$$

with the initial condition

$$u(0, x) = \cos \left( 2\pi \frac{x}{H} \right). \tag{11}$$

We simulate the system with ETDRK4 Cox & Matthews (2002), a different multi-stage PDE solver, under the fine time step $\delta t = 0.02$ for 3000 seconds. The coarse time step for training is $\Delta t = 100\delta t$. We take the time-series data in the range $1000 \leqslant t \leqslant 3000$ for training. This single trajectory is split into 10 consecutive trajectories.

We use a uniform grid with size $n = 48$, and set the hidden layer dimension to 256 and 1024 for NeurTISC and NeurVec, respectively. We train the two models with a time window $L_{\text{train}}$, and use a larger window $L_{\text{test}}$ when making predictions on the unseen data. The comparison of prediction errors is given in Table 6. Spacial-temporal plots for the predicted results of different models are

| $(L_{\text{train}}, L_{\text{test}})$ | $(10, 20)$ | $(20, 50)$ |
|---|---|---|
| ETDRK4 no correction | 9.960e-2 | 7.421e-1 |
| NeurVec | 4.413e-3 | 4.119e-1 |
| NeurTISC (Ours) | **1.835e-3** | **1.462e-2** |

Table 6: Comparison of prediction accuracy on KS equation.

shown in Figure 6. The results show that both ETDRK4 without correction and NeurVec fail to predict the merge of the two attractors near the end while NeurTISC correctly predicts the trajectory of the attractors for 50 future time steps even though the model was trained using a time window of 20 steps.

Finally, we evaluate the effectiveness of the rational activation function on KS system following the same settings used previously. Table 7 shows again that the rational activation function outperform the other two choices.

## 5 CONCLUSION

In this paper we proposed NeurTISC, a new type of neural time integrator based on stage correction. NeurTISC enhances a traditional multi-stage time integrator by using a simple neural network to compensate the errors at each stage; therefore, it allows the use of large step sizes to accelerate the simulation of dynamical systems, including pure ODEs and ODEs converted from PDEs after spacial discretization. We show that the stage-by-stage correction approach can effectively account

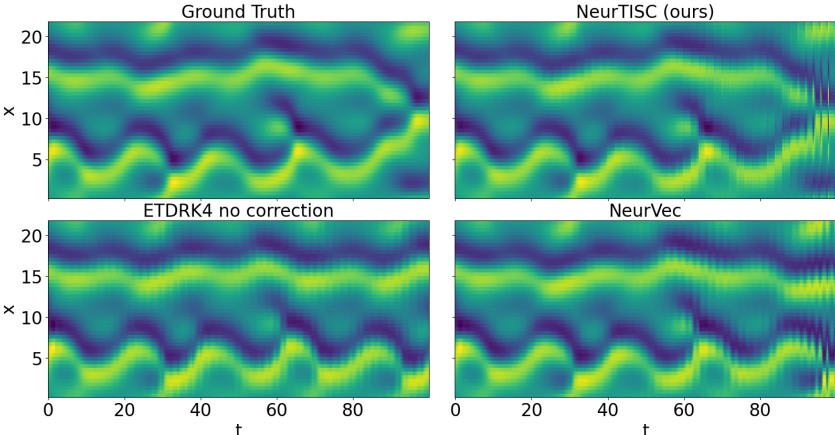

Figure 6: Comparison of path predictions between plain ETDRK4, ETDRK4 with NeurVec integrator, and ETDRK4 with NeurTISC integrator. Note the substantial difference in the predicted solutions in the end.

| $(L_{\text{train}}, L_{\text{test}})$ | (10, 20) | (20, 50) |
|---|---|---|
| GeLU | 2.481e-3 | 1.923e-1 |
| tanh | 1.330e-2 | 3.987e-1 |
| rational | **1.835e-3** | **1.462e-2** |

Table 7: Comparison of prediction accuracy on KS equation using NeurTISC with different activations.

for the distinct error characteristics at each stage of the Runge-Kutta methods and mitigate the error accumulation problem. Using a variety of complex dynamical systems, we demonstrated that NeurTISC achieves at least one order of magnitude better accuracy when compared with the existing method NeurVec, and it can predict the system states for a long time horizon whereas classical integration methods and NeurVec may fail due to instability. We show that NeurTISC does not rely on complex neural network architectures or large number of parameters. Two fully connected layers and a rational activation function suffice to handle the challenging tasks in our experiments. We have also evaluated the effectiveness of the trainable rational activation function through extensive ablation studies. We conclude by emphasizing that hybrid methods like NeurTISC serve as an enhancement to the numerical solvers in dynamical system simulations, ultimately drawing on the strength of physics-based simulations and the speed of ML models.

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
