# OpenReview forum: "Neural Time Integrator with Stage Correction"
_ICLR.cc/2025/Conference — Submitted to ICLR 2025_

### Official Review · Reviewer_5iUq · 2024-11-01

**Soundness:** 2
**Presentation:** 1
**Contribution:** 2
**Rating:** 3
**Confidence:** 4

**Summary:**

This paper proposes a neural time integrator NeurTISC based on stage correction. Unlike NeurVec, NeurTISC compensates for the errors at each stage. NeurTISC achieves better results on three different equations.

**Strengths:**

By correcting errors at each stage, NeurTISC can predict more accurately and more stably than NeurVec and traditional numerical methods.

**Weaknesses:**

1. Note the indentation at the beginning of the paragraph.
2. Many of the quotes require parentheses: Use parentheses in citations when the author’s name is not directly mentioned in the sentence. This helps clearly separate citation information from the main content.
3. The lack of related works. There are quite a lot of other works focusing on introducing neural networks to accelerate traditional methods. [1, 2]
4. Only two time windows $L$ are considered in the Viscous Burgers Equation and Kuramoto–Sivashinsky Equation. Too few types of $L$ will make the results not convincing enough, and it is difficult to reflect the changing trend of the results of different methods as $L$ increases.
5. It will be more convincing to consider and compare the methods on different coarse time step,like multiply by 2 and divide by 2 based on the current time step.
6. Experiments on the accelerated ratio regarding the time of the traditional method and NeurTISC achieving the same accuracy can help demonstrate the effectiveness of NeurTISC. You may reduce the time step of the traditional numerical method or increase the time step of NeurTISC to achieve similar accuracy.
7. l146: the letter after commas should be lowercase: In this section, we...
8. l202: average the loss

[1] Bar-Sinai, Yohai, et al. "Learning data-driven discretizations for partial differential equations." Proceedings of the National Academy of Sciences 116.31 (2019): 15344-15349.
[2] Greenfeld, Daniel, et al. "Learning to optimize multigrid PDE solvers." International Conference on Machine Learning. PMLR, 2019.

**Questions:**

1. l122: what is $a_{ij}$ and $b_i$?
2. l208: what is $\hat{u}$?
3. In the experiments, what kind of metric is used in the loss function?

---

### Official Review · Reviewer_BWF1 · 2024-11-02

**Soundness:** 2
**Presentation:** 2
**Contribution:** 3
**Rating:** 5
**Confidence:** 3

**Summary:**

This paper proposes a new type of neural time integrator based on a stage correction strategy (NeurTISC) to address the challenges of balancing accuracy and computational efficiency in dynamical system simulations. In order to achieve that, the authors introduce a novel integrator with stage corrections, inspired by methods like Runge-Kutta, to correct errors at each stage and allow for larger time steps while maintaining stability and accuracy. The algorithmic contribution is demonstrated with a suite of numerical experiments including elastic pendulum, viscous burgers equation, and viscous burgers equation.

**Strengths:**

(1)	The method proposed is easy to use and effective.
(2)	The related works are adequately stated.
(3)	The method proposed is well demonstrated which makes it relatively understandable.

**Weaknesses:**

(1)The paper does not provide mathematical proof of the convergence stability of NeurTISC.
(2) The numerical experiments are not sufficient to demonstrate the superiority of NeurTISC.
(3) The paper is imprecise and unpolished. The format of the paper needs to be checked for better readability. Some paragraphs begin with an indentation while some do not, which makes your paper look messy. Make sure the format of the paper is aligned with the rules of ICLR. In addition, more information should be included in the figures to make them more understandable. For example, you just mark the horizontal axis as ‘x’ in figure 3, which is very ambiguous. I advise you to use a more understandable word to replace ‘x’ or explain what the meaning of ‘x’ is in the caption of the figure. (4)There is a statement in the conclusion that “therefore, it allows the use of large step sizes to accelerate the simulation of dynamical systems, including pure ODEs and ODEs converted from PDEs after spacial discretization.”  However, the authors do not perform any experiment to compare the number of iteration steps or the computing times between different methods.

**Questions:**

(1) Will the stage-by-stage correction approach you introduc affect the stability of the integrator and make it difficult to converge? You should provide a mathematical proof of the stability of your method or at least cite relevant papers which give a proof.
(2) In your numerical simulations, you just compare NeurTISC with NeurVec and ETDRK4. Why don’t you provide more numerical simulations of other relevant methods? I think it will not consume much time.

---

### Official Review · Reviewer_L54K · 2024-11-03

**Soundness:** 1
**Presentation:** 3
**Contribution:** 1
**Rating:** 3
**Confidence:** 4

**Summary:**

In this paper, the authors propose numerical integration methods for differential equations in which Runge-Kutta methods are modified by neural networks. According to the authors, the proposed numerical method provides more reliable numerical results than purely machine-learning approaches. In particular, in the proposed method, the numerical values at each stage are modified by a neural network, thereby improving the accuracy.

**Strengths:**

Unlike operator learning and PINNs, the proposed method yields modifications of Runge-Kutta methods. Hence numerical solutions obtained by this approach may be able to accurately approximate solutions of differential equations.

**Weaknesses:**

I believe that this paper has several weaknesses.
- In the method proposed in this paper, no constraints on neural networks are imposed. Therefore, numerical methods obtained by this approach is not guaranteed to approximate solutions of the differential equation(i.e., the order of accuracy of the numerical methods is decreased to 0.) This fact disrupts the authors' claim that their method is based on numerical integrators and, hence, highly reliable. For example, for the Heun method described on page 3, $\hat{k}_1 + \hat{k}_2$ must approximate $2F(u)$ to define a method with at least first order accuracy. To this end, a certain condition is required for $NN_1$ and $NN_2$;  for example, if NN_1+NN_2=0, then $\hat{k}_1 + \hat{k}_2$ will approximate $2F(u)$.
- As far as I understand, this method requires the neural network to be re-trained whenever the step size is changed. This is not practical.
- The authors discuss the computational time below (6). It is stated that the proposed method is faster when $\varepsilon$ is small, but in what cases is this expected? Evaluating neural networks requires matrix operations, which require a certain amount of computation.

**Questions:**

My questions are about the practical aspects of the proposed method. Under what conditions can we expect this method to be practical?

---

### Official Review · Reviewer_GmWM · 2024-11-03

**Soundness:** 2
**Presentation:** 3
**Contribution:** 1
**Rating:** 3
**Confidence:** 5

**Summary:**

This paper combines use of the neural networks with conventional time integration methods, e.g. Heun’s method (a second-order Runge-Kutta scheme) to speed up the time integration while maintaining accuracy and stability. The conventional method advances in time with much larger values, and the error of such inaccurate marching is compensated by the use of NNs. While this method has been explored in other papers, e.g. NeurVec, the authors propose use of a network for each "stage" of RK scheme (stage-by-stage correction), observing that various stages have different magnitude, frequency and error properties. They apply this method to both ODEs and PDEs, for the latter they use march in time method (first they discretize the spatial part and then they use NeurTISC for time integration).

**Strengths:**

Kuramoto–Sivashinsky in chaotic regime is a good choice to show the strength of method.
The idea is explained clearly and the paper is well-written.

**Weaknesses:**

Use of Neural networks for time integrator is really now a mature field of its own. This paper does not succeed in positioning itself in the large body of work in this domain. For instance, state-of-the-art is Neural ODE (NODE) which is not mentioned in this paper. There are also other method like hierarchical methods, e.g. Hierarchical Deep Learning of Multiscale Differential Equation Time-Steppers by Liu et al, where a very different architecture is used to achieve the same results by authors.
Also other advanced methods also address other issues that this method is not able to fix; for example large dimensionality of PDEs can be handled by methods like NIF (Neural Implicit Flow: a mesh-agnostic dimensionality reduction paradigm of spatio-temporal data by Pan et al) or CROM (CROM: Continuous Reduced-Order Modeling of PDEs Using Implicit Neural Representations by Chen et al) that not only can take care of large dimensionality of the dynamical system to be integrated, but also provide a continue in time manifold of solution, meaning at each arbitrary instant the solution can be evaluated by the neural network.
The contribution is very incremental, even compared to NeroVec. Multi-frequency of stages can still be captured by a single NN at the last stage in theory, say by using more advanced RNN or autoregressive models.
There is no theoretical analysis to show guarantees and it is mostly based on experimental results which are not the most complicated. problems arise in dynamical systems.

**Questions:**

The introduction part could be revised; For example authors method data-driven and physics-informed; but there could be hybrid version of the two even without referring to conventional methods. What do authors think of this?
How large time step for RK could be chosen for this method? What is the bound after which this method may fail? There should be some analysis on such a choice.
The NN architecture is very simple; which is not necessarily something negative, as it can help with the fast inference evaluation. But have authors also considered my complicated architectures for the correction part?

---

### Meta-Review · Area_Chair_Sd1z · 2024-12-16

**Metareview:**

This paper explores the integration of neural networks with multi-stage Runge-Kutta (RK) methods to accelerate time integration of dynamical systems. Unlike prior work such as NeurVec, which applies a neural network correction after completing all RK stages in an integration step, this paper introduces a stage-by-stage correction, using a separate neural network for each stage. This approach aims to address stage-specific variations in magnitude, frequency, and error properties. The method is evaluated on several ODEs and PDEs, showing improved integration speed and reasonable accuracy.

Despite these results, as reviewers noted, the contribution of this work is quite incremental. The paper does not provide a theoretical explanation for how the proposed method outperforms NeurVec, nor does it include computational efficiency comparisons, such as the number of iterations required to meet a specific error threshold. Furthermore, the method’s practicality is very limited, as the network must be retrained whenever the step size changes.

Given these limitations, the paper is not ready for publication in its current state.

**Additional Comments On Reviewer Discussion:**

The authors did not respond to the reviewers’ comments during the discussion period, and no updates were provided.

---

### Decision · Program_Chairs · 2025-01-22

Reject